# The Role of Consecutive Plasma Copeptin Levels in the Screening of Delayed Cerebral Ischemia in Poor-Grade Subarachnoid Hemorrhage

**DOI:** 10.3390/life11040274

**Published:** 2021-03-25

**Authors:** Jong Kook Rhim, Dong Hyuk Youn, Bong Jun Kim, Youngmi Kim, Sungeun Kim, Heung Cheol Kim, Jin Pyeong Jeon

**Affiliations:** 1Department of Neurosurgery, Jeju National University College of Medicine, Jeju 63243, Korea; pedi-neur@daum.net; 2Institute of New Frontier Stroke Research, Hallym University College of Medicine, Chuncheon 24252, Korea; zk61326@naver.com (D.H.Y.); luckykbj@naver.com (B.J.K.); kym8389@hanmail.net (Y.K.); 3EMS Situation Management Center, Seoul Emergency Operation Center, Seoul Metropolitan Fire & Disaster Headquarters, Seoul 04628, Korea; kimsueu1@hanmail.net; 4Department of Radioilogy, Hallym University College of Medicine, Chuncheon 24252, Korea; khc@hallym.or.kr; 5Genetic and Research Inc., Chuncheon 24252, Korea; 6Department of Neurosurgery, Hallym University College of Medicine, Chuncheon 24252, Korea

**Keywords:** subarachnoid hemorrhage, copeptin, delayed cerebral ischemia, vasospasm

## Abstract

The prognostic value of copeptin in subarachnoid hemorrhage (SAH) has been reported, but the prognosis was largely affected by the initial clinical severity. Thus, the previous studies are not very useful in predicting delayed cerebral ischemia (DCI) in poor-grade SAH patients. Here, we first investigated the feasibility of predicting DCI in poor-grade SAH based on consecutive measurements of plasma copeptin. We measured copeptin levels of 86 patients on days 1, 3, 5, 7, 9, 11, and 13 using ELISA. The primary outcome was the association between consecutive copeptin levels and DCI development. The secondary outcomes were comparison of copeptin with C-reactive protein (CRP) in predicting DCI. Additionally, we compared the prognostic value of transcranial Doppler ultrasonography (TCD) with copeptin using TCD alone to predict DCI. Increased copeptin (OR = 1.022, 95% CI: 1.008–1.037) and modified Fisher scale IV (OR = 2.841; 95% CI: 0.998–8.084) were closely related to DCI. Consecutive plasma copeptin measurements showed significant differences between DCI and non-DCI groups (*p* < 0.001). Higher CRP and DCI appeared to show a correlation, but it was not statistically significant. Analysis of copeptin changes with TCD appeared to predict DCI better than TCD alone with AUCROC differences of 0.072. Consecutive measurements of plasma copeptin levels facilitate the screening of DCI in poor-grade SAH patients.

## 1. Introduction

Delayed cerebral ischemia (DCI) is a clinical condition caused by ischemia resulting in secondary or delayed neurological deterioration in patients with aneurysmal subarachnoid hemorrhage (SAH) [1,2]. DCI requires early diagnosis and prompt treatment to expect a favorable prognosis [3]. Compared to good-grade SAH, it is difficult to detect DCI early in patients with poor-grade SAH because they usually receive intubation and mechanical ventilation or therapeutic coma for increased intracranial pressure (IICP) and neurological changes are difficult to diagnose. Transcranial Doppler ultrasonography (TCD) has a high sensitivity of 90% (95% confidence interval [CI]: 77–96%) and a negative predictive value of 92% (95% CI: 83–96%) in detecting DCI due to arterial vasospasm [1]. However, the accuracy of the test depends on various factors such as the angle between the sound beam and the direction of cerebral flow, flow velocity, ultrasonic frequency, and is limited by poor bone window greater than 10% and operator-dependency [3,4]. Accordingly, in addition to TCD, a sensitive and robust plasma marker is necessary to predict DCI early in patients with poor-grade SAH who are at an elevated risk of DCI and poor neurologic outcomes.

Arginine vasopressin (AVP) is a cerebral vasoconstrictor, which acts on the V1 receptor [5]. Copeptin is a 39-aminoacid glycopeptide located at the C-terminal region of pre-provasopressin (pre-proAVP). Pre-proAVP comprises a signal peptide, AVP, neurophysin II, and copeptin [6,7]. Although the function is largely unknown, copeptin is regarded as a surrogate marker for vasopressin release due to the instability of vasopressin [8]. Plasma levels of copeptin at admission are correlated with the diagnosis of SAH compared with the healthy control group and neurologic outcomes [8,9,10,11,12]. However, in actual clinical practice, additional studies are needed for the following reasons in the field of SAH. First, since there is a computed tomography (CT) that can be taken quickly in most of the clinical practices, it is not meaningful to compare SAH with other cerebrovascular diseases or normal subjects using copeptin level. Second, the analysis of copeptin in SAH patients including good- and poor-grade SAH together is less meaningful in future studies, since copeptin is already associated with SAH severity, arterial vasospasm, and outcomes. Third, most studies analyzed single measurement of plasma copeptin at admission or within 24 h after SAH onset, which may be more appropriate in reflecting early brain injury following SAH than in predicting DCI. Fourth, consecutive measurements of copeptin improved diagnosis and risk stratification in patients with acute chest pain [13]. Zissimopoulos et al. [14] also reported that gradual decreases in copeptin were associated with good prognosis. Therefore, we investigated the role of consecutive measurements of plasma copeptin in predicting DCI in patients with poor-grade SAH, which are difficult to diagnose during general neurological examination.

## 2. Materials and Methods

### 2.1. Study Population

We prospectively collected data at two university hospitals between May 2015 and November 2020 [15,16,17]. Inclusion criteria were: (1) spontaneous SAH; (2) adult patients who were over 18 years old; (3) poor-grade SAH defined as grade 4 or 5 under the Hunt and Hess grading system at admission; (4) treatment within six hours after symptom onset; and (5) no previous history of SAH [18]. The exclusion criteria were: (1) traumatic or infectious aneurysms; (2) good-grade SAH defined as Hunt and Hess grade 1, 2 and 3; (3) spontaneous SAH concomitant with other vascular diseases such as moyamoya disease, arteriovenous malformation or arteriovenous fistula; (4) patients with unknown onset time; and (5) patients who were diagnosed with sepsis and disseminated intravascular coagulation [19].

### 2.2. Treatment Protocol

All patients with reduced consciousness who visited the emergency room were immediately evaluated via CT. A diagnosis of SAH was promptly followed by cerebral angiography. Coil embolization was attempted first in the case of poor-grade SAH patients, followed by external ventricular drainage (EVD) if warranted by thick intraventricular hemorrhage and acute hydrocephalous. Additional craniotomy and removal of hematoma to control IICP was performed as needed. Continuous lumbar drainage of cerebrospinal fluid (CSF) was maintained in the neurointensive care unit in every SAH patient for one week after ictus. Nimodipine (20 μg/kg/hour; Samjin Pharmaceutical, Seoul, Korea) was administered intravenously after the procedure to prevent vasospasm [20].

### 2.3. Detection and Monitoring of DCI

The diagnosis of DCI varies depending on the ability of patients to undergo neurological examination. Patients who were capable of undergoing neurological examination were evaluated using the following diagnostic criteria: (1) new-onset neurological symptoms such as motor weakness, aphasia, and sensory change; (2) decrease in the level of consciousness by more than two points on the Glasgow Coma Scale score; (3) persistence of fluctuating symptoms for more than one hour; (4) exclusion of other causes that may contribute to neurological changes such as re-bleeding, acute hydrocephalus, seizures or electrolyte imbalances; (5) cerebral infarction identified on CT or MRI after excluding procedural complications; and (6) severe angiographic vasospasm, defined as >50% decrease in vessel diameter, in the follow-up CTA or MRA compared to the initial imaging test [21,22,23,24,25]. In the case of patients whose neurological changes were difficult to detect due to sedation with a mechanical ventilator, the DCI diagnosis was based on TCD and imaging tests. TCD was performed daily. Additional CTA or MRA was performed to confirm the degree of vasospasm if severe vasospasm was suspected in TCD, with a mean flow velocity higher than 200 cm/s in the middle cerebral artery (MCA) or 85 cm/s in the basilar artery (BA) [26,27]. Intra-arterial chemical angioplasty using nimodipine was immediately performed 1–2 times daily for 1–7 days depending on the patient’s condition after it was confirmed by imaging tests. Even if the TCD failed to reveal significant increase in cerebral flow velocity, CTA was performed periodically on days 3, 7, and 14 after ictus to observe changes in vasospasm, which begins on day 3, peaks on day 7, and decreases after day 14 [21,28].

### 2.4. Study Outcomes

The primary outcome was prediction of DCI in poor-grade SAH patients depending on consecutive measurements of plasma copeptin levels. The secondary outcomes were: (1) comparison of copeptin with C-reactive protein (CRP), which is closely related to the prognosis of SAH and can be easily used in the hospitals [29] and (2) the increase in the prognostic value of DCI for the analysis of TCD based on plasma copeptin compared with the TCD result alone. Medical records and radiological findings were reviewed independently by the two investigators (JKL and HCK). Disagreements were resolved by the third investigator (JPJ) (Appendix A). Any complications occurring during the procedure such as re-bleeding, infarction, and dissection were deemed to have a significant effect on patient’s prognosis, and those who died within three days were excluded from the final analysis after discussion. This study was approved by the Institutional Review Board (No. 2016-3, 2017-9 and 2018-6) and written informed consent was received from the patients or their relatives.

### 2.5. Blood Sampling and Copeptin Measurement

Venous blood samples were obtained consecutively every two days starting from day 1 of admission until day 13 after ictus in the neurointensive care unit. The blood samples were directly transferred into serum separator tubes and allowed to clot overnight at 4 °C, before centrifugation at 1000× *g* for 15 min. Plasma was derived from whole blood ethylenediaminetetraacetic acid specimens. In brief, the whole blood was centrifuged at 2000 x g for 15 min. Plasma sample was separated, aliquoted, and frozen at −80 °C until assayed. Quantification of plasma copeptin level was measured using ELISA assay (CUSABIO, Wuhan, China) for the detection of horseradish peroxidase (HRP) with 3,3′,5,5′-tetramethylbenzidine (TMB) chromogenic substrates. TMB substrates can be oxidized by HRP and yield the blue color products. Upon adding the stop solution (sulfuric acid in this kit), the color changed from blue to yellow with a maximum absorbance at 450 nm.

### 2.6. Statistical Analysis

Continuous variables are expressed as the mean ± standard deviation (SD). A chi-square or Student’s t-test was used. The degree of agreement between the two investigators was calculated using the k test (Appendix A) [30]. Univariate analysis of factors associated with DCI was carried out and binary logistic regression analysis was further performed to confirm the statistical independence of the variables with p values less than 0.20 [31]. Repeated measures ANOVA, Mauchly’s test of sphericity, and a Greenhouse–Geisser correction were used to evaluate the differences in consecutive measurements of copeptin and CRP according to DCI [20]. Comparative analyses of the plasma levels were presented as mean and 95% CI [32]. A receiver operator characteristic (ROC) curve was generated to determine the optimal cut-off changes, defined as the extent of increase compared to the test result on day one, for predicting DCI. The results of DCI prediction involving both TCD and plasma copeptin compared to the results of TCD alone were determined using the area under ROC curve (AUROC) [33]. Statistical analyses were performed using SPSS V.25 (SPSS, Chicago, IL, USA) and MedCalc (www.medcalc.org, accessed on 5 December 2020) with statistical significance indicated at *p* < 0.05.

## 3. Results

### 3.1. Patients’ Clinical Characteristics

A total of 102 poor-grade SAH patients were initially enrolled. After exclusion, 86 patients were finally included in the analysis (Figure 1) including 56 patients who underwent simple coiling or stent-assisted coil embolization (SACE) and 30 patients treated with additional EVD or craniotomy with hematoma removal after coil embolization. During the follow-up period, 36 patients (41.9%) were diagnosed with DCI. The Cohen’s kappa for DCI diagnosis was 0.880, indicating almost perfect agreement (Appendix A). The differences in the number of clinical characteristics such as female gender, age, hypertension, diabetes mellitus, hyperlipidemia, and smoking did not differ significantly between the DCI and non-DCI groups. Modified Fisher scale IV involving the initial CT was more frequently observed in the DCI (n = 16, 44.4%) than in non-DCI (n = 10, 20.0%), whereas aneurysm location and size did not differ significantly (Table 1). The mean copeptin level was more pronounced in the DCI (295.4 ± 39.4 pg/mL) than in the non-DCI group (263.0 ± 37.1 pg/mL). The mean CRP level tended to increase in poor-grade SAH patients with DCI than in those without DCI, but it was not statistically significant (*p* = 0.085). Additional craniotomy and hematoma removal following endovascular coil embolization was performed in the DCI group (n = 3, 8.3%) more than in the non-DCI group (n = 1, 2.0%; *p* = 0.169).

Binary logistic regression analysis revealed that the mean level of copeptin was the most significant risk factor for DCI development (OR = 1.022, 95% CI: 1.008–1.037). Modified Fisher scale IV on initial CT marginally increased the risk of DCI (OR = 2.841; 95% CI: 0.998–8.084). Other factors such as age, mean CRP and craniotomy and hematoma removal did not reach statistical significance (Table 2).

### 3.2. Consecutive Plasma Copeptin Measurement for DCI Prediction

Plasma copeptin levels were measured on days 1, 3, 5, 7, 9, 11, and 13 after ictus. Consecutive plasma copeptin levels in poor-grade SAH patients with or without DCI are described in Figure 2. The plasma copeptin level increased and peaked on days 7 and 9 after ictus and decreased thereafter. Consecutive plasma copeptin levels were found to be significant between the DCI and non-DCI groups (*p* < 0.001). More specifically, there was no significant difference in copeptin levels until day 5 (286.4 ± 46.1 pg/mL in the DCI and 269.4 ± 47.5 pg/mL in the non-DCI group), but there was a significant difference in the copeptin level after day 7 (Figure 2A). We also compared the consecutive levels of CRP between DCI and non-DCI groups (Figure 2B). The CRP levels measured on day 7 in the DCI group were significantly higher than in the non-DCI group (DCI, 38.1 ± 18.2 mg/L; non-DCI, 29.5 ± 12.4 mg/L; *p* < 0.11); however, the difference in values measured at different periods was not significant.

Based on the rate of increase in plasma copeptin compared with the first-day results as a reference value, we drew an ROC curve (AUC = 0.647). An increase in plasma copeptin by >73% yielded the most favorable outcome with a sensitivity of 66.67% (95% CI: 49.0–81.4%) and a specificity of 66.00% (95% CI: 51.2–78.8%) (Appendix A).

### 3.3. Copeptin with TCD vs. TCD Alone in DCI Prediction

We further evaluated the accuracy of DCI prediction based on changes in plasma copeptin with TCD and TCD alone in patients with poor-grade SAH. Among them, eleven patients (12.8%) were excluded from the comparison due to a poor bone window of TCD. Accordingly, 75 patients were enrolled in the final analysis. As shown in Figure 3, plasma copeptin changes with TCD appeared to be correlated with DCI prediction better than TCD alone with a difference between AUROC areas of 0.072 (95% CI: −0.008 to 0.154; *p* = 0.078).

## 4. Discussion

Copeptin, a 39-amino acid glycopeptide, has been investigated as a direct marker of AVP secretion in various cerebrovascular diseases as well as myocardial infarction, pulmonary diseases, shock, and sepsis [34]. Aksu et al. [9] compared copeptin levels among patients with cerebral infarction, intracranial hemorrhage, SAH, and healthy volunteers in the emergency room. Copeptin levels were significantly increased in patients with cerebrovascular diseases compared with healthy volunteers, but did not differ significantly among patients with infarction, hemorrhage or SAH. Zou et al. [8] reported that initial copeptin levels at admission were related to SAH severity, and mortality rate and poor outcomes were increased by 6% and 9%, respectively, with every 1 pmol/L increase in plasma copeptin level. Zheng et al. [12] also showed that SAH patients with symptomatic vasospasm carried higher copeptin levels than those without symptomatic vasospasm. The copeptin concentration was strongly correlated with the World Federation of Neurological Surgeons subarachnoid hemorrhage scale (WFNS) score or the National Institutes of Health Stroke Scale (NIHSS), suggesting copeptin as a robust indicator of neurological outcomes following SAH. Previous studies of SAH simply correlated copeptin levels measured once and neurologic outcomes or cerebral vasospasm (Table 3). However, most analyses were performed without determining the initial clinical severity. Not surprisingly, patients with high copeptin levels manifested poor-grade SAH at the time of admission, suggesting a high possibility of poor clinical outcomes, underscoring the need to increase the survival rate of patients with poor-grade SAH in the actual neurological intensive care unit. Copeptin levels reflect even mild-to-moderate stress levels [34], suggesting dynamic secretion and requiring serial monitoring during DCI. Dynamic changes in copeptin during the follow-up were associated with prognosis, although the results were based on 32 SAH patients [14]. Fernandez et al. [35] showed varying CSF levels of copeptin in DCI (n = 10) and non-DCI (n = 6) patients: the copeptin levels in DCI patients continued to rise between days 6 and 10 after ictus, whereas copeptin levels in non-DCI patients peaked on days 7 or 8 and decreased thereafter. Thus, the copeptin levels measured at admission or within 24 h after ictus cannot accurately predict DCI. Therefore, we focused on the role of consecutive measurements of plasma copeptin as reliable biomarkers for predicting DCI in poor-grade SAH. Our study demonstrated that consecutive measurement of plasma copeptin facilitated risk stratification of DCI in poor-grade SAH patients.

The mechanism of elevated copeptin in DCI development is still not fully understood. Increased AVP release, co-synthesized with copeptin in the hypothalamus and secreted into the neurohypophysis [8], was associated with acute vasospasm in rodent models. Administration of intracisternal AVP antiserum prior to SAH prevented acute vasospasm [36]. Kagawa et al. [37] reported that treatment with a V1 receptor antagonist of AVP significantly decreased the cortical swelling by diminishing water permeability in the brain. The initial surge in intracranial pressure during acute SAH also disrupts the hypothalamic-pituitary-adrenal axis [38] by triggering sodium imbalance via abnormal release of ADH related to DCI. Pituitary deficiency can occur in up to a third of SAH patients [38]. In particular, the difference in copeptin concentrations between DCI and non-DCI patients was statistically significant based on the degree of hyponatremia [35]. Therefore, copeptin may represent a surrogate marker in the hypothalamus, which reflects AVP release in brain injury after SAH.

Early detection of DCI along with subsequent aggressive treatments is required to obtain favorable outcomes in poor-grade SAH [21]. Clinically, the diagnosis of DCI is important to rule out other causes that contribute to neurologic decline such as re-bleeding, seizures, hydrocephalus, infection, and electrolyte imbalances [21]. Therefore, clinical studies have been conducted to predict DCI early using blood tests. Romero et al. [29] measured CRP daily until 10 days after SAH and reported that higher CRP levels reflected hemodynamic changes. Progressive increase in CRP was observed until three days after ictus, followed by gradual decrease. SAH patients presenting with Hunt and Hess grade ≥ 3 exhibited higher CRP level than those with grade < 3 [29]. Hurth et al. [19] serially measured CRP and D-dimer levels on days 1, 4, 9, 14 and at discharge to evaluate their association with DCI occurrence. They found an increased CRP level in cases of severe vasospasm and increased DCI incidence was correlated with higher D-dimer and Fisher grade IV at admission. Despite multiple blood tests performed several times in previous studies, the results simply established that good or poor prognosis was largely affected by the initial clinical severity. Thus, previous studies are not very useful in predicting the development of DCI in poor-grade SAH patients. Our study demonstrated that elevated plasma copeptin level was a significant risk factor for DCI. In addition to the current TCD results, the analysis of copeptin with TCD tended to increase the accuracy of the prediction.

The study has some limitations. First, we performed coil embolization initially to treat poor-grade SAH. However, other institutions may have different treatment options. In addition, Mees et al. [39] reported that DCI was more common after surgical clipping than coil embolization, although the impact of DCI on neurological outcomes did not differ significantly. Accordingly, the study results may not be appropriate for DCI prediction in patients who underwent surgical clipping in the first place. Second, DCI can develop due to cortical spreading depolarization and microthrombosis as well as severe angiographic vasospasm. It may be easier to differentiate the diseases contributing to DCI via multimodality monitoring of the brain. However, in Korea, it is difficult to monitor cerebral metabolism, perfusion, and oxygenation via multimodality monitoring as it is not covered by insurance benefits. Thus it is possible that DCI due to other causes may be missed. Third, we focused on consecutive changes in plasma copeptin level among various biomarkers related to the occurrence of DCI. Zheng et al. [12] investigated the prognostic significance of copeptin compared with glial fibrillary astrocyte protein, S100B, myelin basic protein, and neuron-specific enolase. Although all the biomarkers mentioned earlier showed significant differences in symptomatic vasospasm, the plasma copeptin level only improved the predictive performance of WFNS score. Based on our results as a reference standard, consecutive changes in plasma copeptin can be used to predict DCI in poor-grade SAH. In particular, the plasma copeptin levels on day seven after SAH ictus clearly differ depending on the DCI development. However, DCI can also occur early after SAH. In our study, six patients experienced DCI within five days after SAH (Appendix A). Early increase in copeptin level was more likely to induce DCI in the early period after SAH. Nevertheless, a biomarker study is needed to improve the accuracy of DCI prediction immediately after SAH.

In conclusion, elevated copeptin is a risk factor for DCI development in patients with poor-grade SAH. Consecutive measurements of plasma copeptin levels can be used to screen patients with poor-grade SAH who are more likely to develop DCI. Early detection of DCI using serial plasma copeptin can lead to prompt treatment, thus a favorable prognosis could be expected. Further studies are needed to verify the usefulness of copeptin in clinical settings.

## Figures and Tables

**Figure 1 life-11-00274-f001:**
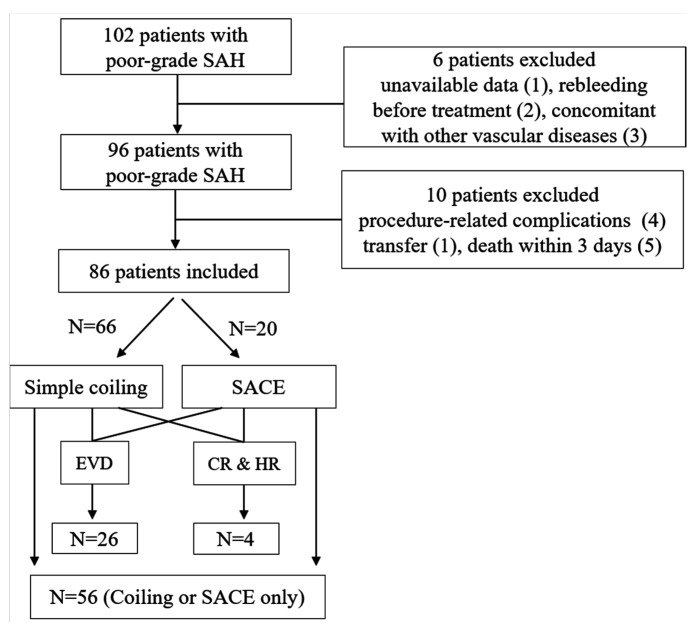
Flow diagram of the study. CR and HR—craniotomy and hematoma removal; EVD—external ventricular drainage; SACE—stent-assisted coil embolization.

**Figure 2 life-11-00274-f002:**
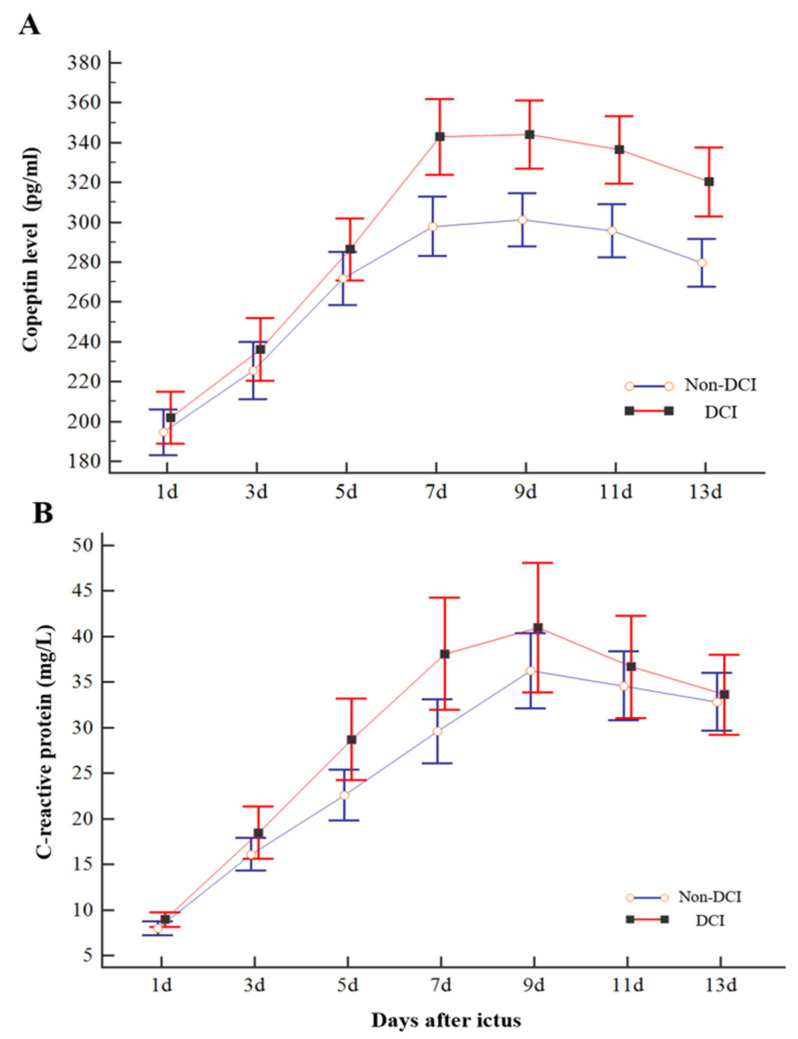
Changes in consecutive copeptin (**A**) and C-reactive protein (CRP) (**B**) levels of patients with poor-grade SAH according to DCI. DCI patients showed a significant increase in copeptin level compared with non-DCI patients on days 7, 9, 11, and 13. However, the difference in CRP measured concurrently did not differ significantly compared to copeptin except for the day 7 results. The bar indicates the mean and 95% confidence interval.

**Figure 3 life-11-00274-f003:**
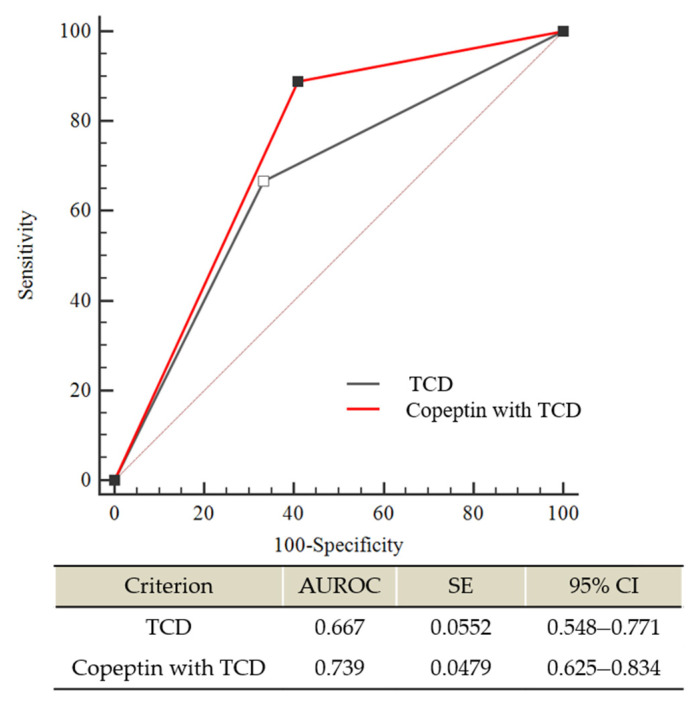
Comparison of receiver operating characteristic (ROC) curves between transcranial doppler ultrasonography (TCD) alone and plasma copeptin changes combined with TCD for predicting severe vasospasm, which contributes to DCI in poor-grade SAH. The difference between AUROC curves was 0.072 (95% CI: −0.008 to 0.154); *p* = 0.0782). AUROC—Area under the ROC curve; CI—confidence interval; SE—standard error.

**Table 1 life-11-00274-t001:** Differences in clinical, radiological, and laboratory parameters: Delayed cerebral ischemia (DCI) vs. non-DCI patients with poor-grade subarachnoid hemorrhage (SAH).

Variables	Non-DCI (n = 50)	DCI (n = 36)	*p*-Value
**Clinical characteristics**			
Female	22 (44.0%)	19 (52.8%)	0.421
Age, years	63.1 ± 8.7	59.3 ± 11.8	0.086
Hypertension	16 (32.0%)	10 (27.8%)	0.674
Diabetes mellitus	8 (16.0%)	6 (16.7%)	0.934
Hyperlipidemia	11 (22.0%)	7 (19.4%)	0.774
Smoking	11 (22.0%)	7 (19.4%)	0.774
**Radiological findings**			
Anterior location	43 (86.0%)	30 (83.3%)	0.733
Size (mm)	5.3 ± 1.2	5.6 ± 1.5	0.372
Modified Fisher scale IV	10 (20.0%)	16 (44.4%)	0.015
**Laboratory results**			
Hemoglobin (g/dL)	11.1 ± 1.1	11.2 ± 1.4	0.760
SaO_2_ (%)	94.7 ± 1.6	94.2 ± 2.6	0.330
Copeptin (pg/mL)	263.0 ± 37.1	295.4 ± 39.4	<0.001
C-reactive protein (mg/L) *	25.7 ± 8.3	29.4 ± 11.2	0.085
**Treatment**			
Simple coiling	37 (74.0%)	29 (80.6%)	0.478
Extraventricular drainage	13 (26.0%)	13 (36.1%)	0.314
Craniotomy and hematoma removal	1 (2.0%)	3 (8.3%)	0.169

Data show the numbers of subjects expressing discrete and categorical variables and mean ± standard deviation. * The normal value of C-reactive protein (CRP) was 0–5 mg/L.

**Table 2 life-11-00274-t002:** Results of binary logistic regression analysis of DCI prediction in poor-grade SAH.

Variables	Odds Ratio	95% Confidence Interval	*p*-Value
Age	0.979	0.932–1.028	0.388
Modified Fisher scale IV	2.841	0.998–8.084	0.050
Copeptin	1.022	1.008–1.037	0.002
C-reactive protein	1.027	0.972–1.085	0.347
Craniotomy and hematoma removal	1.943	0.117–32.367	0.643

**Table 3 life-11-00274-t003:** Summary of studies investigating plasma copeptin levels of SAH patients.

Study, Year	Sample Size	Blood Collection	Detection Method	Number of Blood Collection	Relevance to Copeptin
Zhu 2011 [11]	303	<24 h	ELISA	Single	Outcome and cerebral vasospasm
Fung 2013 [10]	18	Admission	Immunoassay	Single	Severity and prognosis
Zissimopoulos 2015 [14]	32	-	Immunoassay	Several	Severity
Aksu 2016 [9]	29	Admission	ELISA	Single	Comparison with other brain diseases *
Zheng 2017 [12]	105	<24 h	ELISA	Single	Outcome and symptomatic vasospasm
Zuo 2019 [8]	243	Admission	ELISA	Single	Outcome
Present 2020	86 **	Once every two days	ELISA	7	Prediction of DCI in poor-grade SAH

* indicates cerebral infarction and intracranial hemorrhage. ** includes only poor-grade SAH without good-grade SAH. DCI—delayed cerebral ischemia; ELISA—enzyme-linked immunosorbent assay; SAH—subarachnoid hemorrhage.

## Data Availability

Not applicable.

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
