# Peer review of "The Role of Consecutive Plasma Copeptin Levels in the Screening of Delayed Cerebral Ischemia in Poor-Grade Subarachnoid Hemorrhage"

_life, 2021, doi:10.3390/life11040274_

Round 1

Reviewer 1 Report

The idea of the study is clear, data collection and analysis adequate, conclusions are logical. Although the presentation of the study is confusing. Location of the “Materials and Methods” section is unusual and relocating it up to the results section would make the understanding of the text better. 

Author Response

As per your recommendation, we have relocated the Result and Methods section according to the format of the Life journal.

Reviewer 2 Report

The authors aim to assess the predictive value of Copeptin plasma levels after aneursymal subarachnoid hemorrhage. The reviwer finds this approach highly interesting, while the statistics need a serious redo.

Figure 1: 96-6 = 90 (not 86). Obviously, at least 4 other patients have been excluded from the study. What were the reasons to exclude them? Please have the numbers corrected.

The results from the statistical analysis should be interpreted very cautiously: After repeated mesurements ANOVA and Mauchly's test for sphericity, the epsilon value of 0.058 that has resulted from the Huynh-Feldt correction indicates a strong violation of sphericity. (BTW: The manuscript does not clearly indicate whether the authors actually conducted a Huynh-Feldt or a lower-bound correction. Please clarify.) Considering the large number of individuals included in this study and the obvious departure from sphericity, the reviewer suggests to conduct a MANOVA instead of an ANOVA with results that are very hard to interpret.

The statistics as presented here render this manuscript not suitable for further consideration at the current stage.

Author Response

Comment 1: Figure 1: 96-6 = 90 (not 86). Obviously, at least 4 other patients have been excluded from the study. What were the reasons to exclude them? Please have the numbers corrected.

Answer: As per your comments and after careful revision, there seems to be an error in describing the correct number of patients. We have reviewed the data and revised the number of the patients in the flow diagram as follows:

Comment 2: The results from the statistical analysis should be interpreted very cautiously: After repeated mesurements ANOVA and Mauchly's test for sphericity, the epsilon value of 0.058 that has resulted from the Huynh-Feldt correction indicates a strong violation of sphericity. (BTW: The manuscript does not clearly indicate whether the authors actually conducted a Huynh-Feldt or a lower-bound correction. Please clarify.

Answer: Based on your comments and consultation with a statistics expert, we have reviewed the data again and described appropriately in the Method and Result section as follows:

Repeated measures ANOVA, Mauchly’s test of sphericity, and a Greenhouse-Geisser correction were used to evaluate the differences in consecutive measurements of copeptin and CRP according to DCI (page 4, line 139-140).

Consecutive plasma copeptin levels were identified to be significant between DCI and non-DCI groups (page 6, line 1).

Regarding the last comments on the epsilon value of 0.058 that has resulted from the Huynh-Feldt correction, there seems to be an error in the interpretation and description. After consultation with a statistical expert, the epsilon value was found to be 0.584 using a Greenhouse-Geisser correction. To avoid misunderstanding, we have deleted the decription from the manuscript.

Comment 3: Considering the large number of individuals included in this study and the obvious departure from sphericity, the reviewer suggests to conduct a MANOVA instead of an ANOVA with results that are very hard to interpret.

Answer: Thank you for the providing valuable comments on our manuscript. We also agree to some of the reviewer’s comments. In this study, we performed repeated measures ANOVA by repeatedly measuring the result for the same individual over time. The data measured in this way are characterized by being correlated with each other within the variable, and have the advantage of increasing the accuracy of the study as it can detect small fluctuations compared to simple ANOVA. In the case of MANOVA, this method is used to distinguish the vector of group means with two or more dependent variables. In this case, it is advantageous to find the combined difference compared with simple ANOVA method. Surely, it may be easier to interpret the statistical results using MANOVA than our present analysis method as per the reviewer’s comment. However, after consulting with a statistical expert, the analysis was conducted using the existing method, and the results and the description of the methods are presented in a simpler way.

Reviewer 3 Report

Delayed cerebral ischemia (DCI) is one of the leading causes of poor outcome after subarachnoid hemorrhage (SAH). It is characterized by a neurological deficit generally observed between days 3 and 14 after hemorrhage (de Oliveira Manoel et al. Critical Care, 2016:20:21). Although clinical and radiological features of SAH may help in detecting patients who can develop DCI, there are no definitive serum-based biomarkers which can detect DCI in SAH patients. In the current study, the authors have measured consecutive copeptin levels in patients with poor-grade SAH to predict DCI. They have observed a statistically significant increase (7 days onwards) in copeptin levels in SAH-patients with DCI, compared to the non-DCI group. An increase in CRP levels is associated with occurrence of DCI after SAH, and authors have also tested the correlation between copeptin and CRP as a secondary outcome to predict DCI. Early detection and timely intervention are required to reverse DCI as soon as possible before the ischemic process leads to infarction. This study will be valuable in predicting DCI in patients with SAH.

Minor Comments:

  1. Page 1, line 38: The phrase “DCI is the most important preventable cause of mortality and neurological outcome” is a verbatim from the ref [3], authors might want to reword it.
  2. Page 2, line 57: The sentence “First, the need to distinguish between SAH and other cerebrovascular diseases or normal subjects is not higher due to imaging tests” is not clear.
  3. Page 2, line 82: Provide long form of CRP, at its first use. May be in abstract too.
  4. Page 3, line 88: Please provide long forms (Abbreviation list) for CR, HR, SACE, EVD) in the figure legend.
  5. Page 8, line 231: Add concluding remarks on the benefits of early detection of DCI in SAH patients.
  6. Page 8, line 249: CSF was drained from all patients with SAH. Did authors measure copeptin levels CSF too?
  7. Page 9, line 287: Authors are saying the blood samples were collected in serum separator tubes. When blood samples are clotted, and centrifuged, the supernatants will be serum samples. This study says plasma analysis. Please clarify. Moreover, the blood samples are clotted just by leaving them at room temperature for about 30 minutes after the collection. It is not clear why authors stored then overnight at 4 degree.
  8. Page 8, line 289: In this sentence authors are saying that they measured optical density at 450 nm. Which samples they are referring to? Are they antibody treated plasma/serum samples during ELISA assay? Please specify.

Author Response

Comments 1: Page 1, line 38: The phrase “DCI is the most important preventable cause of mortality and neurological outcome” is a verbatim from the ref [3], authors might want to reword it.

Answer: As per your recommendation, we have revised the sentence as follows:

DCI requires early diagnosis and prompt treatment to expect a favorable prognosis.

Comments 2: Page 2, line 57: The sentence “First, the need to distinguish between SAH and other cerebrovascular diseases or normal subjects is not higher due to imaging tests” is not clear.

Answer: First, since there is a computed tomography that can be taken quickly in most of the clinical practices, it is not meaningful to compare SAH with other cerebrovascular diseases or normal subjects using copeptin level.

Comments 3: Page 2, line 82: Provide long form of CRP, at its first use. May be in abstract too.

Answer: As per your recommendation, we have added description about the C-reactive protein (CRP) in the abstract and its first use in the manuscript.

Comments 4: Page 3, line 88: Please provide long forms (Abbreviation list) for CR, HR, SACE, EVD) in the figure legend.

Answer: As per your recommendation, abbreviation description has been added to Figure 1.

Comments 5: Page 8, line 231: Add concluding remarks on the benefits of early detection of DCI in SAH patients.

Answer: As per your recommendation, we have added the concluding remark as follows:

Early detection of DCI using serial plasma copeptin can lead to prompt treatment, thus a favorable prognosis could be expected. Further studies are needed to verify the usefulness of copeptin in clinical settings.

Comments 6: Page 8, line 249: CSF was drained from all patients with SAH. Did authors measure copeptin levels CSF too?

Answer: In some patients, copeptin in the CSF was measured along with plasma copeptin. We are working on this as a different research topic, especially regarding electrolyte abnormalities following SAH.

Comments 7: Page 9, line 287: Authors are saying the blood samples were collected in serum separator tubes. When blood samples are clotted, and centrifuged, the supernatants will be serum samples. This study says plasma analysis. Please clarify. Moreover, the blood samples are clotted just by leaving them at room temperature for about 30 minutes after the collection. It is not clear why authors stored then overnight at 4 degree.

Answer: We would like to apologize for the mistakes. As per your comments, we have revised the methods as follows:

Plasma was derived from whole blood ethylenediaminetetraacetic acid specimens. In brief, the whole blood was centrifuged at 2000 x g for 15 minutes. Plasma sample was separated, aliquoted, and frozen at -80°C until assayed.

Comments 8: Page 8, line 289: In this sentence authors are saying that they measured optical density at 450 nm. Which samples they are referring to? Are they antibody treated plasma/serum samples during ELISA assay? Please specify.

Answer: Quantification of plasma copeptin level was measured using ELISA Kit for the detection of horseradish peroxidase (HRP) with 3,3',5,5'-tetramethylbenzidine (TMB) chromogenic substrates. TMB substrates can be oxidized by HRP and yield the blue color products. Upon adding the stop solution (sulfuric acid in this kit), the color changes from blue to yellow with a maximum absorbance at 450 nm.

Reviewer 4 Report

Proper research seems correct to me. I suggest adapting it to the magazine template, and putting the methods before the results.a

Author Response

(The authors gave the same response as above.)

Round 2

Reviewer 2 Report

The concerns of the reviewer have been properly addressed.